Methods

# Reference-free transcriptome exploration reveals novel RNAs for prostate cancer diagnosis

Marina Pinskaya[1],*[iD], Zohra Saci[1],*, Mélina Gallopin[2], Marc Gabriel[1], Ha TN Nguyen[2,3], Virginie Firlej[4,5], Marc Descrimes[1], Audrey Rapinat[6], David Gentien[6][iD], Alexandre de la Taille[4,5,7], Arturo Londoño-Vallejo[8][iD], Yves Allory[9], Daniel Gautheret[2][iD], Antonin Morillon[1][iD]

**The use of RNA-sequencing technologies held a promise of improved diagnostic tools based on comprehensive transcript sets. However, mining human transcriptome data for disease biomarkers in clinical specimens are restricted by the limited power of conventional reference-based protocols relying on unique and annotated transcripts. Here, we implemented a blind reference-free computational protocol, DE-kupl, to infer yet unreferenced RNA variations from total stranded RNA-sequencing datasets of tissue origin. As a bench test, this protocol was powered for detection of RNA subsequences embedded into putative long noncoding (lnc)RNAs expressed in prostate cancer. Through filtering of 1,179 candidates, we defined 21 lncRNAs that were further validated by NanoString for robust tumor-specific expression in 144 tissue specimens. Predictive modeling yielded a restricted probe panel enabling more than 90% of true-positive detections of cancer in an independent The Cancer Genome Atlas cohort. Remarkably, this clinical signature made of only nine unannotated lncRNAs largely outperformed PCA3, the only used prostate cancer lncRNA biomarker, in detection of high-risk tumors. This modular workflow is highly sensitive and can be applied to any pathology or clinical application.**

## Introduction

RNA sequencing (RNA-seq) has revolutionized our knowledge of human transcriptome and has been implemented as a pivot technique in clinical applications for discovery of RNA-based biomarkers allowing disease diagnosis, prognosis and therapy follow-up. However, most biomarker discovery pipelines are blind to uncharacterized RNA molecules because they rely on the alignment of uniquely mapped reads to annotated references of the human transcriptome, which are far from complete (Deveson et al, 2018; Uszczynska-Ratajczak et al, 2018; Morillon & Gautheret, 2019). Indeed, unspliced variants, rare mRNA isoforms, RNA hybrids originating from *trans*-splicing or genome rearrangements, unannotated intergenic or antisense noncoding RNAs, mobile elements, or viral genome insertions would be systematically missed. A recent approach to RNA-seq data analysis, DE-kupl, combines k-mer decomposition and differential expression analysis to discover transcript variations yet unreferenced in the human transcriptome (Audoux et al, 2017). Applied to poly(A)+ RNA-seq datasets of in vitro cell system, DE-kupl unveiled a large number of RNA subsequences embedded into novel long noncoding (lnc)RNAs. These transcripts of more than 200 nucleotides in length transcribed by RNA polymerase II from intergenic, intronic, or antisense noncoding genomic locations constitute a prevalent class of human genes. Some lncRNAs are now recognized as precisely regulated stand-alone molecules participating in the control of fundamental cellular processes (Quinn & Chang, 2015; Jarroux et al, 2017). They show aberrant and specific expression in various cancers and other diseases promoting them as biomarkers, therapeutic molecules and drug targets (Van Grembergen et al, 2016; Leucci, 2018). Importantly, some lncRNAs can be robustly detected in biological fluids (blood and urine) as circulating molecules or encapsulated into extracellular vesicles, hence, raising an attractive possibility of their usage as biomarkers in non-invasive clinical tests (Wang et al, 2014; Silva et al, 2015; Deng et al, 2017; Wang et al, 2018; Zhao et al, 2018). The only example of a lncRNA-based biomarker so far introduced in clinical practice of prostate cancer (PCa) is the PCA3 lncRNA (de Kok et al, 2002). PCA3 is transcribed antisense to the tumor suppressor *PRUNE2* gene and

---

[1]ncRNA, Epigenetic and Genome Fluidity, Université Paris Sciences & Lettres (PSL), Sorbonne Université, Centre National de la Recherche Scientifique (CNRS), Institut Curie, Research Center, Paris, France [2]Institute for Integrative Biology of the Cell, Commissariat à l'Energie Atomique, CNRS, Université Paris-Sud, Université Paris-Saclay, Gif sur Yvette, France [3]Thuyloi University, Hanoi, Vietnam [4]Université Paris-Est Créteil, Créteil, France [5]Institut National de la Santé et de la Recherche Médicale, U955, Equipe 7, Créteil, France [6]Translational Research Department, Genomics Platform, Institut Curie, Université PSL, Paris, France [7]Assistance Publique – Hôpitaux de Paris, Hôpital Henri Mondor, Département d'Urologie, Créteil, France [8]Telomeres and Cancer, Université PSL, Sorbonne Université, CNRS, Institut Curie, Research Center, Paris, France [9]Compartimentation et Dynamique Cellulaire, Université PSL, Sorbonne Université, CNRS, Institut Curie, Research Center, Paris, France

Correspondence: antonin.morillon@curie.fr; daniel.gautheret@u-psud.fr
Zohra Saci's present address is CHU Sainte-Justine Research Centre, University of Montreal, Montreal, Quebec, Canada
*Marina Pinskaya and Zohra Saci contributed equally to this work

promotes its pre-mRNA editing and degradation (Salameh et al, 2015). Being overexpressed in 95% of PCa cases, PCA3 is detected in urine and helps diagnosis providing, in addition to other clinical tests, more accurate metrics regarding repeated biopsies (Groskopf et al, 2006; Galasso et al, 2010). However, it remains inaccurate in discrimination between low- and high-risk tumors because its expression may dramatically decrease in aggressive PCa cases tempering its systematic usage (Loeb & Partin, 2011; Alshalalfa et al, 2017).

Since PCA3 discovery and the development of RNA-seq technologies, the PCa transcriptome has been extensively explored by The Cancer Genome Atlas (TCGA) consortium and others to identify numerous PCa-associated lncRNAs (PCAT family) such as PCAT1, PCAT7, or PCAT114/SChLAP1 (Prensner et al, 2014; Iyer et al, 2015). However, none of them has been yet introduced into clinical practice because of the variable expression incidence, as for SChLAP1 detected in 25% of PCa cases presenting metastatic traits (Prensner et al, 2013), or low specificity, as PCAT1 or PCAT7, thus infringing their clinical value. Additional efforts are required for more accurate and exhaustive RNA identification, as well as more rigorous validations of clinical potency through independent RNA measurement technologies and clinical cohorts. Regardless a large number of transcriptomic studies and variety of clinical samples analyzed, discovery of RNA-based biomarkers from publicly available RNA-seq datasets is still limited at two levels: (i) most experimental setups are based on poly(A) selected, unstranded cDNA sequencing, and (ii) computational analyses are generally focused on annotated genes and full-length RNA assemblies. This impedes the detection of low and poorly polyadenylated RNAs but also partially degraded RNAs from formalin-fixed paraffin-embedded tissues or other clinical samples (Zhao et al, 2014; Zhao et al, 2018). In addition, non-stranded RNA-seq reads counting is less accurate at 5′ RNA ends or even impossible for co-expressed paired sense/antisense transcripts and for yet unannotated RNAs among noncoding, fusion, repeat-derived transcripts (Davila et al, 2016; Audoux et al, 2017).

Here, we propose a conceptually novel exploratory framework combining the total stranded RNA-seq of clinical samples and the reference-free DE-kupl algorithm for discovery of novel tumor-specific transcript variations. As a proof-of-concept, we focused on the least explored, noncoding portion of the genome devoid of annotated protein-coding sequences to build an exhaustive catalog of PCa associated subsequences (contigs) embedded into lncRNA genes. The catalog was further refined through minimal filtering to isolate the subset of contigs with best differential expression features and validate 21 of them by a custom NanoString assay in the extended cohort of 144 prostate specimens. From this, a predictive modeling derived a panel of nine yet unannotated lncRNAs validated for robust expression in an independent TCGA cohort. Importantly, its clinical performance surpassed the PCA3 lncRNA specifically in discrimination of high-risk tumors. The proposed probe-set can be further used for development of a PCa diagnostic test. Moving beyond this point, the proposed computational and experimental platform may serve as a tool for biomarkers discovery of any disease and any clinical task aiming at improved medical care and development of precision medicine approaches.

# Results

## Identification of PCa-specific RNA variants in the *Discovery Set* by DE-kupl

The biomarker discovery workflow included three major phases: discovery, selection, and validation (Fig 1). First, for discovery, we performed a deep total stranded RNA-seq of ribosomal RNA-depleted RNA samples isolated from prostate tissues after radical prostatectomy (*Discovery Set*, PAIR cohort, Table S1). This *Discovery Set* was processed by DE-kupl to identify tumor-specific transcripts. DE-kupl directly queries FASTQ files for subsequences (k-mers) with differential counts/expression (DE) between two conditions (Fig 2A) (Audoux et al, 2017). Overlapping k-mers are then assembled into contigs and, in a final step, mapped to the human genome for annotation. In the aim to focus exclusively on novel, yet unannotated RNA elements, k-mers exactly matching GENCODE-annotated transcripts were masked. We eventually retained contigs within the noncoding regions (antisense to protein-coding or noncoding genes, intergenic) longer than 200 nucleotides and showing adjusted *P*-values below 0.01 to capture the most significant expression changes linked either to new transcriptional or processing events within known or putative lncRNA loci.

With these criteria, we identified 1,179 tumor up-regulated contigs assigned to four main categories according to their mapping features: contiguous (uniquely mapped) contigs (N = 935), splice variants (N = 54), repeats (N = 167), and unmapped contigs (N = 23) (Figs 2B and S1, and Table S2). Among them, 586 contigs were

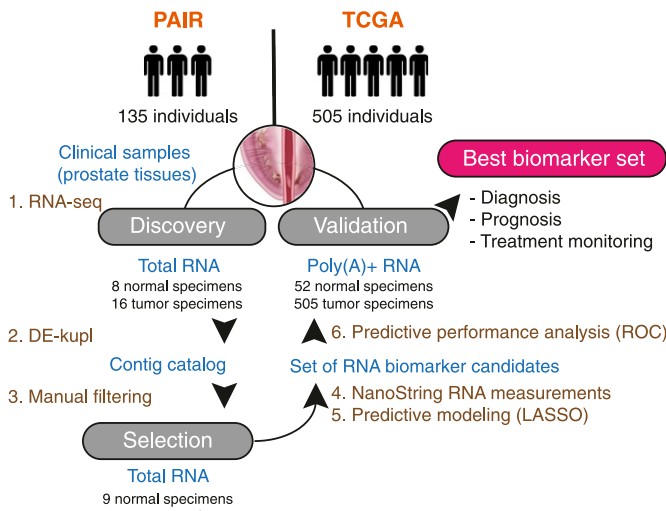

**Figure 1. Experimental and computational workflow for discovery and validation of RNA-based clinical biomarkers.**
Raw total stranded RNA-seq data of a small clinical cohort is processed by DE-kupl to allow comparison of 8 normal against 16 tumor specimens (in this case, formaldehyde-fixed paraffin-embedded tissues from radical prostatectomy) and cataloguing of all differentially expressed RNA variations (contigs). The whole set is filtered according to desired criteria and the top ranked contigs are selected for an independent experimental validation by NanoString in the extended clinical cohort. Finally, predictive modeling infers the best panel of candidate RNAs for validation of its clinical potency in an independent cohort (in this case TCGA).

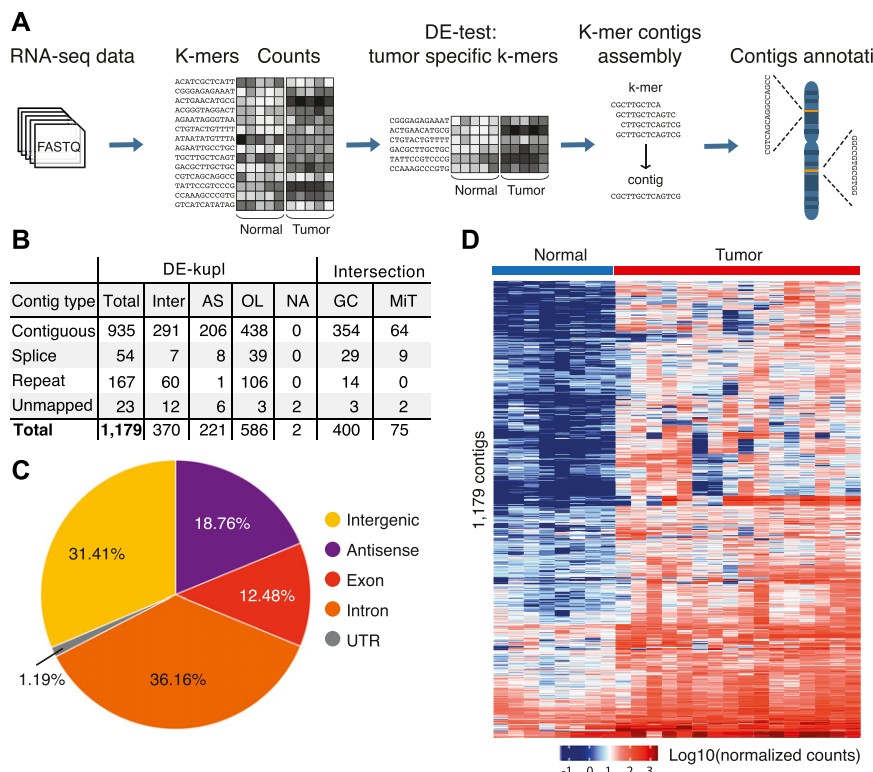

**Figure 2. K-mer decomposition protocol for detection of differentially expressed RNA variants in PCa.**
**(A)** DE-kupl workflow with principle steps of contigs counting, DE-test and filtering, assembly and annotation. **(B)** Catalog of DE-kupl contigs of different subgroups: contiguous—contigs mapped as unique fragments; spliced—contigs mapped as spliced fragments; repeat—multiply mapped contigs; Inter—contigs mapping into intergenic regions; OL—at least one nucleotide overlapping of GENCODE lncRNA annotations; and AS—antisense to a protein-coding or a noncoding gene. Contigs of each subgroup showing 50% sequence overlap with GENCODE v27 (GC)– and MiTranscriptome v2 (MiT)–annotated genes are counted. **(C)** Pie chart of 1,179 contigs distribution across GENCODE-annotated features. **(D)** Unsupervised hierarchical cluster heat map of Log10(normalized counts) of 1,179 contigs assessed in 8 normal and 16 tumor specimens by total stranded RNA-seq of the *Discovery Set*. NA stands for non-annotated in human genome.

embedded into already referenced GENCODE lncRNA genes, but represented new sequence variations or RNA processing events, as PCAT7 (ctg_111158, P6) or CTBP1-AS (ctg_25348, P10). The rest mapped to intergenic noncoding locations (370 contigs) or antisense to referenced protein-coding or noncoding genes (221 contigs) (Fig 2C). Intersection with existing annotations revealed 50% sequence overlap of contigs with 400 (33.93%) GENCODE and 75 (6.36%) MiTranscriptome lncRNA genes (Fig 2B). An unsupervised clustering of prostate specimens based on contigs expression counts allowed proper discrimination of tumor from normal tissues of the *Discovery Set* (Fig 2D).

In conclusion, DE-kupl identified a thousand of PCa-associated RNA variants for the majority embedded into yet unreferenced transcripts which may represent putative novel lncRNAs. This depository was further explored for clinical relevance.

## Naïve assembly of transcription units identifies novel prostate cancer associated lncRNAs

To complement the reference-free protocol, we applied a reference-based protocol to build a catalog of lncRNAs from the same *Discovery Set*. Total RNA-seq produces much more intronic and exon–exon junction reads than poly(A)–selected RNA-seq. This complexity renders laborious in time and machine memory the data analysis by splice graph–based assemblers such as Cufflinks (Hayer et al, 2015; Kukurba & Montgomery, 2015). To bypass this difficulty, we developed a more straightforward lncRNA annotation pipeline, HoLdUp, which identifies transcription units (TUs) based on coverage analysis (Fig 3A). In this workflow, uniquely mapped reads were assembled into TUs and mapped to the GENCODE annotation to extract intergenic

and antisense lncRNAs (see the Materials and Methods section for details). They were further ranked according to their expression level, presence of splice junctions, and existence of matched ESTs. In total, we retained 168,163 TUs with above-threshold expression of 0.2 quartile of mRNA expression (Class 2) and, within this group, the most robust 2,972 TUs with at least one splice junction and one EST (Class 1) (Fig 3B). Globally, newly detected transcripts were as much expressed as GENCODE-annotated lncRNAs but lower than mRNAs (Fig S2A). Only 0.33% of Class 1 lncRNAs were present with at least 50% nucleotide sequence overlap in the recent GENCODE v27 catalog and 43.37% of TUs in the MiTranscriptome lncRNA repertoire; the rest represented putative novel lncRNA genes (Figs 3B and S2B). Of 2,972 TUs, DE analysis retrieved 127 of Class 1 TUs significantly up-regulated in tumor specimens (adjusted *P*-value below 0.01, DESeq), including multiple intergenic transcripts and transcripts antisense to protein-coding genes, such as HDAC9, TPO, and FBXL7 (Table S3 and Fig S2B).

Intersection of DE-kupl contigs with PCa up-regulated HoLdUp TUs (N = 127) and the recent GENCODE lncRNA annotation (N = 206) showed that 687 DE-kupl contigs of 1,179 make part of the stand-alone transcripts. Moreover, up to 85.5% and 96.8% DE-kupl contigs embedded into GENCODE and HoLdUp Class 1 lncRNA genes, respectively, were also detected by DESeq as significantly up-regulated transcripts in the same dataset, when the RNA-seq reads were counted within the entire TU (Figs 3C and S2C). One such example is the contig ctg_23999 (P22) embedded into a novel HoLdUp assembled Class 1 TU antisense to the protein-coding FBXL7 gene (Fig 3D).

In conclusion, the reference-based assembly protocol HoLdUp is complementary to DE-kupl and allows attributing short RNA

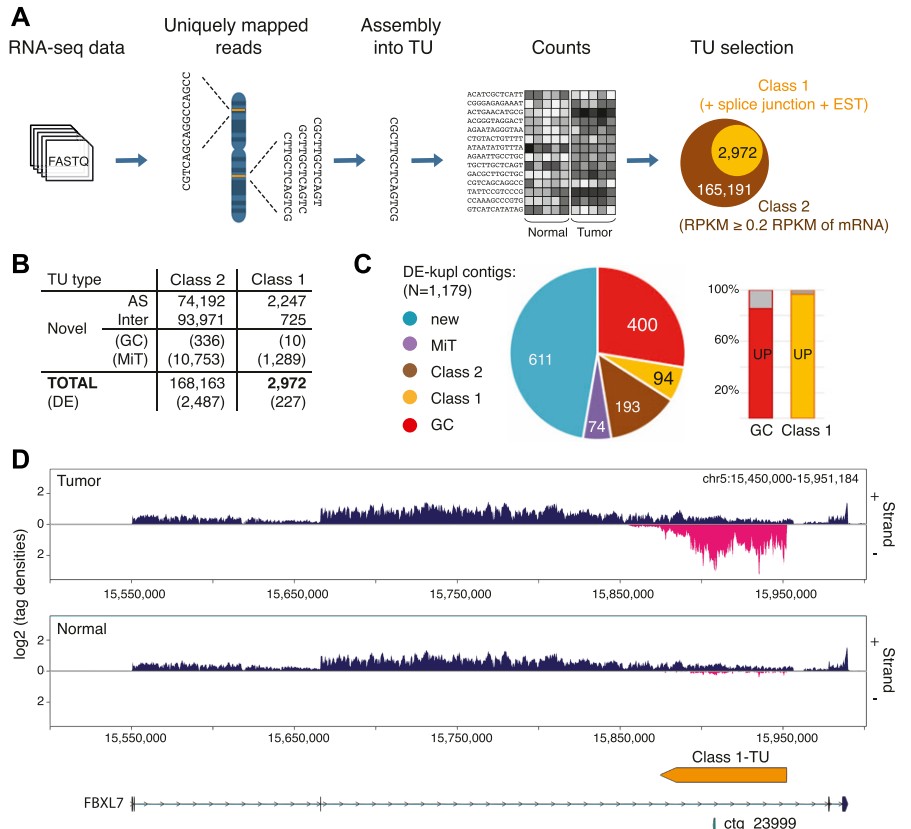

**Figure 3. Reference-based lncRNA discovery from total stranded RNA-seq.**
**(A)** HoLdUp protocol for the ab initio assembly of TUs constituting putative lncRNA genes and their classification into Class 2 and Class 1 TUs according to robustness of detection. **(B)** HoLdUp catalog and TUs overlap with GENCODE v27– (GC) and MiTranscriptome (MiT)–annotated lncRNAs. DE stands for differentially expressed transcripts (DESeq adj. *P*-value < 0.01). **(C)** Pie chart representation of non-exclusive distribution of DE-kupl contigs across different lncRNA annotations: MiTranscriptome (violet), Class 1 (yellow), Class 2 (brown), GENCODE (red), and novel (blue); number of contigs is marked in each section. Proportion of DE-kupl contigs embedded into up-regulated (UP) GENCODE (red) and Class 1 (yellow) lncRNAs is expressed as a histogram. **(D)** VING-generated RNA-seq profiling along plus (+) and minus (−) strands of chr5:15,500,295-15,939,910 in tumor and normal prostate specimens: the GENCODE-annotated protein-coding gene *FBXL7* (blue), antisense DE-kupl contig ctg_23999 (P22), and antisense HoLdUp Class 1-TU (orange). Arrow-lines and rectangles represent introns and exons, respectively. DE, differentially expressed; RPKM, reads per kilo base per million mapped reads; and TU, transcription unit.

subsequences to whole TUs. Nevertheless, DE-kupl was more powerful illuminating much more transcriptomic variations not only within the annotated genes but also within putative new noncoding regions in highly complex and heterogeneous total RNA-seq datasets of clinical origin.

## Selection of a restricted set of 23 PCa RNA contigs showing the highest differential expression

We further leveraged the DE-kupl contig catalog to define a robust PCa signature among putative new lncRNAs using several filters (Fig S3A). Hereafter, we will use the term *signature* to describe the set of contigs or genes selected for their ability to predict a sample status. First, contigs were sorted according to their adjusted *P*-value and, second, were visually selected using the Integrative Genomic Viewer applying the following criteria: (i) when several contigs were present within the same genomic region (5 kb window) the contig with the lowest adjusted *P*-value was retained, (ii) contigs antisense to expressed exons, bidirectional or positioned in close vicinity to other transcribed protein-coding genes were filtered out. We retained several contigs embedded into already annotated PCa associated lncRNA genes, such as *CTBP1-AS* (ctg_25348, P10), *PCAT7* (ctg_111158, P6), and *PCAT1* (ctg_105149, P18), or lncRNAs referenced elsewhere as ctg_104447 (P11) mapped into *LOC283177*, ctg_123090 (P5) into *AC004066.3*, and ctg_73782 (P8) into *LINC01006*. It should be noted that the GENCODE referenced genes enclosing these new subsequences also showed differential expression when counting

on the whole gene annotation (Fig S3B). However, in contrast to DE-kupl ranking, they were not among the strongest hits in the DESeq analysis with exception of PCAT7 (Table S4). This observation points to the fact that through expression counting within the small subsequences, DE-kupl is more resolutive and hence sensitive in the discovery of DE sequences. Visualization of RNA-seq reads and junctions of a region embedding *FBP2* and its antisense *PCAT7* genes revealed a new contig ctg_28650 (P2) downstream of the *PCAT7* annotation and antisense to *FBP2*. The continuous coverage and absence of splice junctions in reads profiling suggest that P2 is enclosed into an extension of the last *PCAT7* exon (Fig S3C and D). This contig was retained in the restricted list as the strongest candidate antisense to *FBP2*, overcoming ctg_111158 (P6) assigned to the *PCAT7* gene itself. Still, additional experiments are required to validate this lncRNA variant, yet absent from the existing PCAT7 annotation.

In total, 23 candidates belonging to contiguous (N = 21), spliced (N = 1), or repeat (N = 1) subgroups of contigs were selected for further validation, all being expressed at least six times more in tumor tissues than in normal prostate (Fig S3E and Tables S2 and S5). Among them, 12 candidates mapped antisense to annotated protein-coding or lncRNA genes and 11 located to intergenic regions. To facilitate further reading, contigs' identity are replaced by probes' identity from P1 to P23 according to increasing *P*-values of DE of the *Discovery Set* (Table S5).

After the manual filtering, we aimed to validate the expression of selected 23 contigs in the extended PAIR cohort of nine normal and

135 tumor specimens (*Selection Set*) (Table S6). This cohort contained one additional specimen for normal tissue and 119 additional tumor specimens. To measure contigs expression, an alternative RNA quantification procedure based on the NanoString nCounter platform for direct enzyme-free multiplex digital RNA measurements was carried out (Fig 4A). In addition to DE-kupl contigs, a probe for PCA3 was used as a benchmark lncRNA. We also measured the expression of six housekeeping genes and selected three lowly expressed mRNAs (GPATCH3, ZNF2, and ZNF346) as custom internal controls for relative quantifications (Table S7 and Fig S4).

The NanoString assay revealed that all DE-kupl contigs were expressed at a lower level than PCA3, but still 21 of 23 contigs were significantly overexpressed (Wilcoxon *P*-value < 0.01) in tumor specimens (Fig 4A and Table S8). Two contigs, intergenic P22 (ctg_119680) and repeat P17 (ctg_36195), did not show significant difference in expression between normal and tumor specimens. Ranking according to *P*-values revealed 12 contigs better than PCA3. Among the top DE contigs were those embedded into *PCAT1* (ctg_105149, P18), *CTBP1-AS* (ctg_25348, P10), and *PCAT7* (ctg_111158, P6) genes, whereas the rest were assigned to novel lncRNAs. Notably, apart from P17 (ctg_36195) and P22 (ctg_119680), expression measurements were consistent between the two technologies, total stranded RNA-seq and NanoString, although the *P*-value ordering was different (Fig S5 and Table S9).

Thus, 21 of 23 contigs were validated in the extended set of RNA specimens using the independent single-molecule measurement technology.

### Validation of contig-based RNA candidates in an independent clinical cohort

Independent validation of DE-kupl contigs was performed using the biggest PCa clinical resource of 557 poly(A)+ RNA-seq datasets, including 52 normal and 505 tumor tissues from radical prostatectomy (TCGA-prostate adenocarcinoma [PRAD] cohort, *Validation Set*) (Fig 1 and Table S10).

The occurrence of sequences representing 23 DE-kupl contigs was measured and compared with PCA3. In total, 16 of 23 DE-kupl contigs had significant support for overexpression in tumor specimens in the TCGA-PRAD cohort (Wilcoxon *P*-value < 0.01, Fold Change [FC] > 2) (Fig 4B and Table S11). Among the best scored candidates, the two novel DE-kupl contigs, P16 (ctg_111348) antisense to *DLX1* and intergenic P1 (ctg_17297), surpassed PCA3 that ranked third. However, important discrepancies were observed between expression counts in poly(A)+ RNA-seq TCGA datasets and NanoString or total RNA-seq PAIR datasets. First, P22 (ctg_119680) was detected as DE in TCGA-PRAD but failed the DE test when measured by NanoString (Figs 4 and S5). Second, the expression of nine DE-kupl contigs were near the base line in the TCGA dataset, including those showing relatively high expression and low *P*-values in the PAIR cohort, such as P14 (ctg_61528) antisense to *TPO* or the intergenic P9 (ctg_9446). Detection of these contigs in TCGA-PRAD was compromised independently of their genomic location (intergenic or antisense) or of the expression level of a sense-paired gene. We hypothesized that it is most likely due to a relatively low RNA-seq coverage and/or to a loss of poorly or non-polyadenylated transcripts during cDNA library preparation in the TCGA experimental setup. Finally, ranking of contigs according to increasing *P*-values was very different between *Selection* and *Validation Sets* highlighting discrepancies between technologies, clinical origins, and cohort sizes.

Regardless all experimental biases, 16 of 23 DE-kupl contigs were validated in the independent clinical cohort as significantly overexpressed in tumors. This cohort was further used for validation of clinical potency of contigs.

### Expression of DE-kupl contigs is independent of tumor risk and recurrence metrics

Several clinical studies have revealed high heterogeneity of expression and low efficiency of the PCA3 biomarker in detection of high-risk tumors, questioning its robustness and reliability in PCa

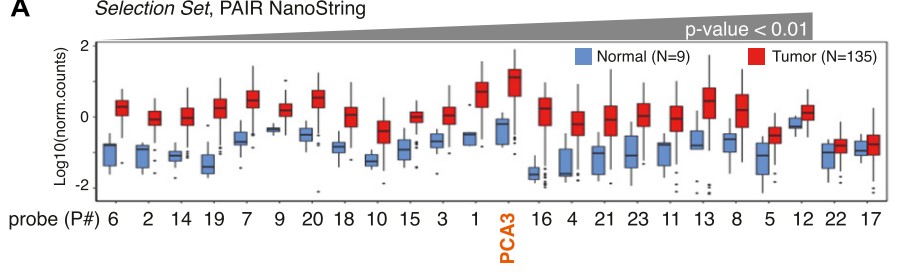

**A** *Selection Set*, PAIR NanoString

**B** *Validation Set*, TCGA-PRAD Poly(A)+ RNA-seq

**Figure 4. Expression of lncRNA subsequences in PAIR and TCGA-PRAD cohorts.**
**(A)** Box-plot of Log10(norm.counts) of PCA3 and 23 DE-kupl contigs in 144 PAIR specimens of the *Selection Set* by NanoString. **(B)** Box-plot of Log10(norm.counts) of PCA3 and 23 DE-kupl contigs in 557 TCGA-PRAD specimens of the *Validation Set* by poly(A)+ unstranded RNA-seq. Normal tissues: in blue, tumor tissues: in red.

diagnostics (Alshalalfa et al, 2017; Fenstermaker et al, 2017). We assessed contig expression in tumors of different clinical metrics. For risk prognosis, the most common metric is a three-group risk stratification system established by D'Amico et al (1998), which takes into account preoperative PSA level, biopsy Gleason Score, and clinical TNM stage. As mentioned above, this scheme is highly debated because of disagreements on the PSA score in relation to PCa over-diagnosis (Carlsson et al, 2012; Loeb et al, 2014). To define a molecular signature independent of PSA, we excluded this criterion and categorized tumor specimens into low-, intermediate-, and high-risk groups uniquely on the basis of Gleason and TNM features, below referred to as naïve indexing (Fig S6A and B). In addition to risk assessment, we also separated specimens in two subgroups depending on the tumor recurrence status (Fig S6B). Then, expression of PCA3 and the 23 DE-kupl contigs were compared for each subgroup of the *Selection Set*.

To evaluate the robustness of contig expression, we ranked probes by decreasing FC for high-risk against low-risk tumors and positive against negative recurrence status (Fig 5). Most contigs showed robust expression independently of the tumor classification. In contrast, the PCA3 level was more disperse with the lower median and mean expression and higher *P*-values in high-risk and recurrence positive specimens (Table S12). While considering only 21 significantly overexpressed contigs, 17 of them outperformed PCA3 in both contrasts (Table S12). Notably, among the best performe were contigs P6 (ctg_111158) and P2 (ctg_28650) both antisense to *FBP2*, P10 (ctg_25348) embedded into CTBP1-AS, as well as the novel P16 (ctg_111348) antisense to *DLX1* and the intergenic P1 (ctg_17297).

In conclusion, most DE-kupl contigs showed robust expression independent of tumor metrics. Hence, even if used alone, they may offer a better clinical potency for PCa diagnosis than PCA3.

### Inferring a multiplex RNA-probe panel and evaluation of its performance in PCa diagnosis

To extract parsimonious probe signature predicting the tumor status, we applied Least Absolute Shrinkage and Selection Operator

(LASSO) logistic regression on the *Selection Set* of 144 PAIR specimens (Ghosh & Chinnaiyan, 2005). First, the initial 21 DE-kupl contigs and PCA3 validated for expression by NanoString were submitted to LASSO to define the best mixed signature comprised of already known and yet unannotated lncRNA probes for discrimination of tumor from normal tissues (Fig S7A). Then, LASSO was performed with the probe subset composed uniquely of contigs assigned to putative novel lncRNAs (N = 15) to infer the best new-lncRNA signature. It resulted in two panels of nine mixed and nine new-lncRNA candidates (Figs 6A and S7B). Retrieved signatures were then used to predict a tumor status in the *Validation Set* of the TCGA-PRAD cohort using a leave-one-out cross-validated boosted logistic regression. To assess the sensitivity of DE-kupl contigs in PCa diagnosis, a predictive accuracy index, area under curve (AUC) of the receiver-operating characteristic (ROC), was calculated for each signature and PCA3 alone in the PAIR (*Selection Set*) and TCGA-PRAD (*Validation Set*) datasets (Figs 6B and S7B). Remarkably, all signatures still hold their predictive capacity in the independent TCGA-PRAD cohort in spite of the important differences in experimental setups between the two studies. Both markedly outperformed PCA3 for tumor detection with AUC of 0.92 for mixed and of 0.91 for new-lncRNA signatures against AUC of 0.73 for PCA3 (Fig 6B and C). In addition, these signatures were much better in predicting high-risk tumors where PCA3 is particularly inaccurate (Fig 6C). Remarkably, the new-lncRNA signature composed uniquely of yet unannotated lncRNA subsequences predicted the tumor status with the same performance as the mixed signature. Logistic regression did not retain PCA3 within the mixed signature set, instead contigs embedded into the well characterized PCAT1 lncRNA and into two already annotated but yet functionally uncharacterized lncRNAs LOC283177 and LINC01006 were present.

We also compared predictive performances of signatures retrieved by the k-mer–based classifier to the one inferred using conventional gene expression counting. Differential expression analysis for GENCODE-annotated genes of the *Discovery Set* retrieved 520 up-regulated genes, protein-coding and noncoding, with adjusted *P*-values lower than 0.05 and a logFC higher than 2

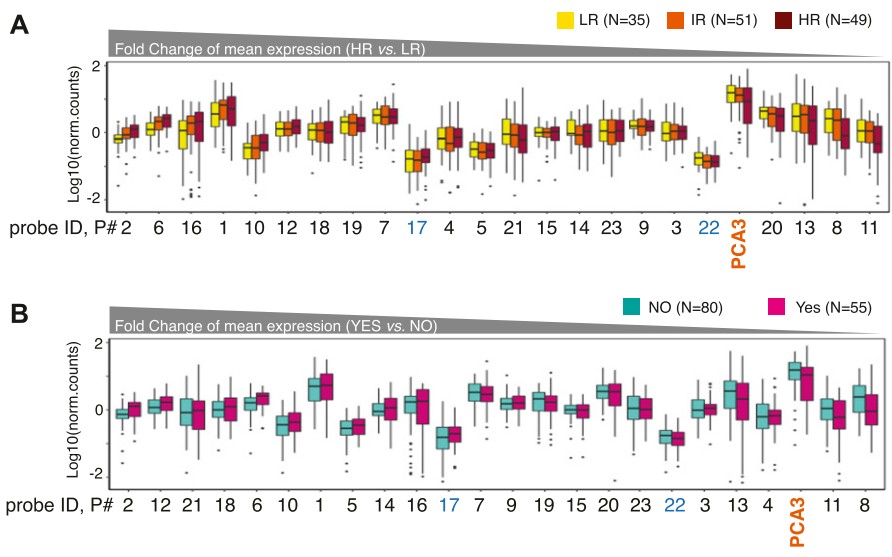

**Figure 5. Expression of lncRNA subsequences in prostate specimens of different clinical metrics in the PAIR cohort (*Selection Set*).**
**(A, B)** Box-plot of Log10(norm.counts) of PCA3 and 23 DE-kupl contigs depending on tumor risk (A) and recurrence status (B) assessed by NanoString. PCA3 is marked in orange, and the contigs showing insignificant expression change between normal and tumor specimens are in blue. Contigs are ordered by the decreasing FC of mean expression in high-risk versus low-risk specimens in the (A) panel and in Yes versus NO recurrence specimens in the (B) panel. HR, high-risk; IR, intermediate-risk; LR, low-risk.

**A**

Signature

| Probe | mixed | new-lnc | contig origin |
|---|---|---|---|
| P8 | ctg_73782 | | LINC01006 |
| P18 | ctg_105149 | | PCAT1 |
| P11 | ctg_104447 | | LOC283177 |
| P1 | ctg_17297 | ctg_17297 | intergenic |
| P2 | ctg_28650 | ctg_28650 | AS to *FBP2* |
| P7 | ctg_117356 | ctg_117356 | AS to *snoU13* |
| P15 | ctg_512 | ctg_512 | AS to *PXDN* |
| P20 | ctg_44030 | ctg_44030 | integenic |
| P23 | ctg_29077 | ctg_29077 | AS to *AC011523.2* |
| P3 | | ctg_57223 | intergenic |
| P12 | | ctg_2815 | intergenic |
| P14 | | ctg_61528 | AS to *TPO* |

**C**

| | PCA3 | mixed | new-lnc |
|---|---|---|---|
| Normal *vs.* Tumor | 0.73±0.05 | 0.92±0.03 | 0.91±0.03 |
| Normal *vs.* HR | 0.69±0.05 | 0.91±0.03 | 0.91±0.03 |
| Normal *vs.* IR | 0.78±0.05 | 0.90±0.05 | 0.90±0.04 |
| Normal *vs.* LR | 0.78±0.11 | 0.92±0.04 | 0.91±0.03 |

**B**

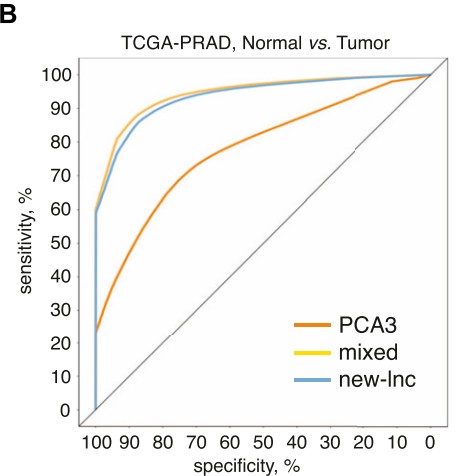

TCGA-PRAD, Normal *vs.* Tumor (sensitivity, % vs. specificity, %)

— PCA3
— mixed
— new-lnc

**Figure 6. Predictive performance of PCA3 and multiplex mixed and new-lncRNA signatures inferred from the LASSO penalized logistic regression.**
**(A)** Multiplex biomarker signatures composed of either known and unannotated RNAs (mixed) or of only unannotated RNAs (new-lnc). **(B)** ROC for the PCa prediction in the TCGA dataset (*Validation Set*) using two signatures and PCA3 alone. **(C)** Mean and SD of AUC computed over 100 samplings of the *Validation Set* for PCA3 and two signatures to classify samples according to their tumor status. AS, antisense; AUC, area under the curve; HR, high-risk; IR, intermediate-risk; LR, low-risk tumors.

(Table S4). These genes were then selected on the *Discovery Set* using LASSO penalized logistic regression to extract a GENCODE whole gene counting based (WGC) signature for further validation (Fig S7B). Given the high dimensional setting (more variables than observations available), we performed the stability selection (Meinshausen & Bühlmann, 2010) and kept the five genes that had a probability of being selected higher than 0.5 on 2,000 samplings of the original dataset. Remarkably, the retrieved subset was composed majorly of noncoding transcripts (four of five), although PCA3 did not pass the selection. Of them, the protein-coding HPN mRNA, the PCAT7 lncRNA, and the GLYATL1P4 pseudogene have been already associated with PCa in other studies (Willard & Koochekpour, 2012; Du et al, 2013; Kim et al, 2019). Notably, the GLYATL1P4 transcript makes part, together with 21 other RNAs, of the Decipher test proposed in clinics to guide timing of radiation therapy after radical prostatectomy in men with high-risk cancer (Alford et al, 2017). The predictive performance in discrimination between normal and tumor specimens of the WGC signature was tested by the ROC analysis on the *Validation Set* and resulted in the mean AUC of 0.91 (Fig S7B). Hence, k-mer based signature discovery method retrieving yet unreferenced RNA subsequence was as powerful as the signature derived from GENCODE-annotated genes. Although the predictive modeling enabled to reach the same performance only from 23 contig probes instead of 520 DE-genes and, remarkably, this was achieved in TCGA-PRAD datasets where contigs expression counting is most likely disfavored considering all aforementioned drawbacks of poly(A)–selected datasets of low coverage.

Discovery of novel RNA signatures with high tumor predictive potential also highlights both the incompleteness of current cancer transcriptome datasets and the biological value of transcript information that can be extracted through different experimental (total stranded RNA-seq and NanoString quantification) and computational (DE-kupl) tools. De-kupl–derived novel signature demonstrated a sensitivity and robustness towards tumor risk prediction surpassing the state of the art for discrimination of prostate cancer. Furthermore, established nine-probe RNA signature was performed not only independently of tumor origin and its clinicopathological characteristics but also of the technology used for RNA measurements.

## Discussion

Molecular biomarker assays are invaluable tools in cancer diagnosis, prognosis and treatment follow-up. Within this scope, sequencing technologies unveiled the pervasiveness and diversity of the human transcriptome, promoting lncRNAs as important cancer signatures (Schmitt & Chang, 2016). These molecules are highly dynamic and reflect cellular states in a sensitive and specific way because of their involvement in genetic and regulatory flows of information. However, the variety of RNA species and high heterogeneity of expression present a challenge for their detection and proper quantification in clinical samples. Predominant microarray and unstranded poly(A)+ RNA-seq–based approaches allowed identification of numerous lncRNAs with tumorigenic function. However, their clinical performance as biomarkers stays rather poor because of the aforementioned RNA features hindering RNA detection, quantification, and clinical validation under conventional experimental setups. Here, we presented an innovative experimental and computational platform that permits discovery of RNA biomarkers of high clinical potency from total stranded RNA-seq datasets of clinical origin.

As a proof-of-concept, we focused on PCa as the only type of cancer using, so far, a lncRNA-based diagnostic test (Progensa). The *Discovery Set* based on comparison of 8 normal with 16 tumor specimens from total RNA-seq datasets was processed by DE-kupl to extract the most significant differentially expressed subsequences in the form of k-mer contigs. Further filtering based on contig length, genomic position, and expression levels powered the pipeline towards the discovery of putative lncRNAs, for the majority, yet unreferenced in the human transcriptome. Then, the catalog of contigs was manually refined and tested for expression using the NanoString single-molecule RNA counting technology in the extended cohort of 144 specimens. Contig expression was next assessed in the independent, publicly available TCGA-PRAD dataset generated by the poly(A)+ unstranded RNA-seq technology. The expression of contigs was systematically compared with that of the benchmark biomarker lncRNA, PCA3. In total, 16 of 23 contigs were validated in both setups but with important differences. Primarily,

RNA measurements were consistent between two different technologies: NanoString and total stranded RNA-seq. In contrast, the TCGA poly(A)+ unstranded datasets revealed weakness and high heterogeneity of contig counts over the selected regions, resulting in unexpectedly low signals even for PCA3, considered as a highly expressed lncRNA. Hence, our results promote the total stranded RNA-seq as a first-choice strategy for discovery of RNA biomarkers from clinical samples and when searching for transcripts others than highly abundant mRNAs. It reflects far more precisely the transcriptomic landscape of clinical samples and, hence, is more advantageous as a *Discovery Set* for development of clinical tests. At the same time, full-length transcript assembly from short-read sequencing is inaccurate, time and computer memory consuming, and this is aggravated by the added complexity of total (ribo-depleted) RNA-seq libraries (Hayer et al, 2015). DE-kupl bypasses this issue by directly extracting from raw data RNA subsequences significantly overexpressed in a defined condition. In PCa tissues, this allowed identification of 1,179 lncRNA-hosted candidates. Further analysis isolated a restrained set of nine contigs either within putative new lncRNAs or mixed annotated and novel lncRNAs allowing PCa diagnosis independently of tumor risk classifications with higher accuracy than the actual PCA3. Remarkably, the best performing mixed signature did not include PCA3, consistent with the low potency of this biomarker in detection of aggressive tumors. Instead, both mixed and new-lncRNA signatures contained contigs embedded into putative novel lncRNA genes. We strongly believe that these signatures can complement the existing clinical tests as lncRNA-based PCA3 (Progensa) or mostly mRNA-based Decipher to improve the accuracy of tumor stratification and clinical decisions for better patient care (Alford et al, 2017). Still, in this study, to compute signature coefficients, sample information (normal or tumor) was used because the extended *Selection* and independent *Validation Sets* used two different technologies for RNA measurements. This precluded us from calculating an objective signature performance. An additional cohort using the same NanoString technology as the *Selection Set* should now be tested to explore the clinical potential of the obtained signature.

In addition to the clinical value, functions of the newly discovered lncRNA variants embedding DE-kupl contigs will be important to explore. Foremost, proper assignment of contigs to stand-alone transcripts is required, and this task can be accomplished computationally through ab initio discovery and assembly of novel transcripts as demonstrated here by HoLdUp or other assemblers, and then through experimental validation at the transcript-specific or transcriptomic level. In the latter case, high-throughput RACE (rapid amplification of cDNA ends) or long-read RNA-seq approaches can be useful. Among others, detailed examination of newly discovered contigs revealed a genomic locus on chromosome 19 transcribed in PCa specimens in both directions into the GENCODE-annotated AC011523.2 lncRNA and a novel, antisense transcript embedding the P23 contig (ctg_29077). Located between *KLK15* and the PSA encoding *KLK3* genes, this region makes part of a super-enhancer annotated in several PCa cell lines (Jiang et al, 2019). Moreover, bidirectionally produced enhancer RNAs from this locus have been shown to regulate the expression of neighboring *KLK3* and *KLK2* genes through Med1-dependent chromatin

looping in several PCa cell lines (Hsieh et al, 2014). Presence of the P23 contig within the mixed and new-lncRNA signatures supports, in addition to the clinical potency, possible regulatory functions of the RNA contigs inferred by DE-kupl. More globally, most DE-kupl contigs within co-transcribed sense–antisense pairs were annotated as super-enhancers in prostate tissues and cell lines or other biosamples, for example, P15 (ctg_512), P7 (ctg_117356), and P4 (ctg_63866) (Jiang et al, 2019). In most cases, their function in gene expression regulation and chromatin configuration has not yet been investigated and experimentally validated, but it is tempting to speculate that defined sense-antisense transcripts may influence a super-enhancer activity and, consequently, may fine-tune the expression of neighboring genes.

In this work, we propose DE-kupl as a tool for discovery of novel disease-associated transcriptomic variations, which can be further explored for biological and clinical relevance. As a pilot project, we oriented the pipeline towards the discovery of novel lncRNAs, but using proper masking and filtering criteria defined by the investigator, other variant transcripts, including single nucleotide variations, novel splice events, gene fusions, circular RNAs, or exogenous viral RNAs, could be probed. The workflow can be applied to any RNA-seq datasets of any clinical origin (tissue, blood, and urine) to generate a probe panel that may be implemented as a multiplex platform for simultaneous detection of RNAs in clinical samples. Moreover, different experimental contrasts (normal versus pathology, low- versus high-risk grade, chemoresistant versus sensitive, etc.) will define the biomarker usage in diagnosis, prognosis, or other clinical applications, hence providing clinicians and researchers with a simple and highly sensitive tool for genomic and personalized medicine.

# Materials and Methods

### Tissue samples

Tumor and normal biopsy specimens were retrospectively collected from prostate cancer patients who provided informed consent and were approved for distribution by the Henri Mondor institutional board (PAIR cohort). Tumor classification in low-, intermediate-, and high-risk prognosis was performed according to Gleason and TNM scores and regardless PSA values (Table S1 and Fig S6B).

### RNA extraction, quantification, and cDNA library production

Total RNA was extracted using the TRizol reagent (Thermo Fisher Scientific), according to the manufacturer's procedure, quantified, and quality-controlled using a 2100 Bioanalyzer (Agilent). RNA samples with RNA Integrity Number (RIN) above six were depleted for ribosomal RNA and converted into cDNA library using a TruSeq Stranded Total Library Preparation kit (Illumina). cDNA libraries were normalized using an Illumina duplex-specific Nuclease protocol before a paired-end sequencing on HiSeq 2500 (Illumina). At least 20× coverage per sample was considered as minimum of unique sequences for further data analysis.

### RNA-seq data

Raw paired-end strand-specific RNA-seq data were generated by our laboratory from ribo-depleted total RNA samples of prostate tissues (8 normal and 16 tumor specimens, Table S1) and can be retrieved from the gene omnibus portal, accession number GSE115414. TCGA prostate cancer poly(A)–selected RNA-seq and corresponding clinical data were obtained from publicly available TCGA dataset (http://cancergenome.nih.gov), 557 inputs in total (52 normal and 505 tumors of high- [N = 240], intermediate- [N = 128], and low-risk [N = 132] groups). Among them, 369 patients showed no tumor recurrence, 108 presented a new tumor event (Table S10).

### Computational workflow for k-mer contigs discovery from total stranded RNA-seq dataset

DE-kupl run was performed from (June 2017) with parameters ctg_length 31, min_recurrence 6, min_recurrence_abundance 5, pvalue_threshold 0.05, lib_type stranded, diff_method DESeq2. K-mer masking was performed against the GENCODE v24 annotation. DE-kupl analysis of the 8 against 16 PAIR RNA-seq prostate libraries yielded 124,809 DE contigs, in total. Contigs were annotated by alignment on the hg19 human genome assembly using the DE-kupl *annotate* procedure. We further selected contigs of size above 200 nucleotides and classified them into four categories (contiguous, repeat, spliced, and unmapped) based on their location and mapping features (Table S2).

### Computational workflow for reference-based ab initio transcripts assembly from total stranded RNA-seq dataset (HoLdUP)

The human genome version hg19 and the GENCODE v14 annotation were used in this study. First, we performed a quality control of all sequencing data by FastQC Babraham Bioinformatics software. Reads were mapped using TopHat 2.0.4, allowing three mismatches and requesting uniquely mapped reads, which were further assembled using the BedTools suite. Overlapping contigs from all libraries were merged, and only contigs supported by at least 10 reads in either library were further assembled in segments if mapped in the same strand and separated by less than 100 nucleotides. We compared the segments with the GENCODE v14 annotation to extract antisense and intergenic TUs longer than 200 nucleotides. To classify lncRNAs, we applied the following criteria: (i) an expression level above 0.2 quartile of mRNA expression in at least one condition per tissue (Class 2); (ii) within this class, all TUs containing at least one TopHat-identified exon–exon junction and at least one spliced EST from UCSC mapped contigs were assigned to Class 1. The whole catalog, the R code, and Data Tables can be downloaded from https://github.com/MorillonLab/HoLDuP_pipeline.

### Overlap between GENCODE, MiTranscriptome, DE-kupl, and HoLdUp catalogues

Intersection between transcripts was counted only in the case of 50% overlap of nucleotide sequence between genomic coordinates of each fragment.

### Differential expression analysis

Read counting was performed on the compiled annotation (GENCODE v27, HoLdUp Class 1 and Class 2) for each sample, using *featureCounts* 1.6.0 with the following parameters: -F "SAF" -p -s 2 -O and the *DESeq R* package (Liao et al, 2014; Love et al, 2014). Only RNAs with adjusted *P*-value below 0.01 were retained as differentially expressed to constitute the prostate tumor signature (Tables S3 and S4). Gene expression counts were normalized using the DESeq2 median of ratio (Anders & Huber, 2010). Scripts are available at https://github.com/MorillonLab/Prostate_additional_scripts.

### NanoString nCounter expression assay

100 ng of total RNA was used for direct digital detection of 29 target transcripts: six housekeeping genes (*RPL11*, *GAPDH*, *NOL7*, *GPATCH3*, *ZNF2*, and *ZNF346*), 23 contigs and the one known PCa-associated lncRNA, PCA3. Each target gene of interest was detected in RNA samples of 144 specimens (9 normal and 135 tumor) of the PAIR cohort (Table S6) on NanoString nCounter V2 using reporter and capture probes of 35- to 50-nucleotide targeting sequences listed in Table S4. Data was normalized through the use of NanoString's intrinsic negative and positive controls according to the normalization approach of the nSolver analysis software (https://www.nanostring.com/products/analysis-software/nsolver) and then contig expression was calculated relative to the average signal of three housekeeping genes (*GPATCH3*, *ZNF2*, and *ZNF346*). Raw and normalized data for each specimen, and mean and fold change expression in normal against tumor samples are presented in Tables S7 and S8.

### Contig expression measurements in TCGA-PRAD datasets

DE-kupl provides representative k-mers for each differentially expressed contig. We converted the TCGA-PRAD FASTQ files to k-mer counts using *Jellyfish count* and counted representative k-mers in each Jellyfish count file using the *Jellyfish query* command (Marçais & Kingsford, 2011). Counts were normalized by total number of reads in corresponding libraries. To determine whether counts of DE-kupl derived representative k-mer were a reliable proxy for evaluating contig expression, we compared representative k-mer counts to average counts from k-mers sampled along each contig. All individual counts were obtained using *Jellyfish Dump* files produced for each TCGA-PRAD library. Sampling was performed as follows: (i) we extracted all k-mers from the contig that were unique in the Ensembl human v91 transcript reference, and (ii) from this list, we sampled 10 regularly spaced k-mers, starting from the first 10% and ending in the last 10% of the list. This sampling procedure was repeated four times for each contig. For the whole TCGA library and each contig, the 10 k-mer counts obtained by Jellyfish were averaged, yielding one average count per sample per library Table S13. Pearson correlation analysis for two DE-kupl contigs P1 and P16 are shown in Fig S8A and B. Jellyfish commands can be retrieved from https://github.com/MorillonLab/Prostate-kmer-signatures.

### RNA-seq data visualization

RNA-seq reads profiling along a locus of interest was performed using our in-house R script VING using one "normal" and one

"tumor" RNA-seq subsets build by random sampling of 10% of reads from each raw data sample (Descrimes et al, 2015). The normal samples were assigned to the group "controls" and the tumor specimens–to the group "cases," with the assumption that the "cases" should have higher values than "controls."

### Unsupervised clustering of prostate specimens

Specimens were ranked based on the Log10(norm.counts) levels of contigs assessed by the NanoString nCounter assay using a ComplexHeatmap R-package (Gu et al, 2016). Scripts are available from GitHub: https://github.com/MorillonLab/Prostate_additional_scripts.

### Variable selection using the LASSO penalized logistic regression and external validation of signatures

Signature inference was performed in R using the normalized *Selection Set* (23 probes in 144 observations) as a variable selection dataset and contigs counts table of the *Validation Set* (23 probes in 557 observations) as an external validation dataset (R Core Team). First, we performed penalized logistic regression using the *glmnet* R package to select probes predicting the tumor status on the *Selection Set* upsampled to correct the imbalance class distribution (9 normal versus 135 tumor specimens) (Friedman et al, 2010). Selection was performed using all probes (signature_mixed including PCA3) or using only new-lncRNA contigs only (signature_new-lnc) (Fig S7). Second, we built predictors using the boosted logistic regression from the *caTools* and *caret* packages (Kuhn, 2008; Tuszynski, 2008). Note that the final gene subsets (signatures) do not have coefficients computed on the *Selection Set* over the *Validation Set* because in contrast to NanoString, the TCGA-PRAD RNA-seq datasets are poly(A)–selected and unstranded. To build the ROC curves, we sampled 100 datasets in two, for training (70%) and testing (30%) preserving the relative ratio of labels in each. We used boosted logistic regression with upsampling, setting the number of boosting iterations to 100 and using leave-one-out cross validation scheme on the training set. After training, we evaluated the predictor on the testing set and repeated the procedure for each one of the 100 training and testing sets described above to obtain an average ROC curve, mean and SD for AUC scores. Contig expression counts in the *Validation Set* (TCGA-PRAD) were obtained as described above using the DE-kupl derived representative k-mer for each contig. Quantifications based on 10 randomly sampled k-mers per contig did not alter predictive performance (Fig S8C). To build a classifier based on the conventional WGC procedure, we used DESeq2 across the GENCODE annotation on the *Discovery Set* and kept only up-regulated genes with adjusted *P*-value lower than 0.05 and Log2FC higher than 2. To perform gene selection on the *Discovery Set*, we used LASSO penalized logistic regression combined with stability selection. Only genes with probability above 0.5 on 2,000 up-regulated samples from the initial dataset were retained. The remaining genes were then used to build ROC curves and compute the mean and SD of the AUC on the *Validation Set* as described above for the DE-kupl-derived representative k-mers. The results file, R codes, and data tables are provided through the GitHub repository: https://github.com/MorillonLab/Prostate-kmer-signatures.

### Data access

Raw paired-end strand-specific RNA-seq data can be retrieved from the gene omnibus portal, accession number GSE115414. TCGA prostate cancer poly(A)–selected RNA-seq and corresponding clinical data can be obtained from TCGA portal (https://www.cancer.gov/tcga).

# Supplementary Information

# Acknowledgements

We deeply thank Dominika Foretek, Maxime Wery, and Alexandre Serero for editorial suggestions; Camille Gautier, Claire Bertrand, and Anna Almeida (Morillon lab, Institut Curie) and Sylvain Baulande for RNA-seq (Next Generation Sequencing platform, Institut Curie); and Cedric Saule and Jeremy Le Coz for the DE-kupl run (Gautheret lab, I2BC). The Cancer Genome Atlas RNA-seq data for prostate adenocarcinoma (PRAD) were downloaded from the dbGaP Web site under authorization granted to D Gautheret (project #13359). Funding: Constitution of the prostate cancer cohort was performed with the financial support from the INCa-Ligue-ARC PAIR program to Y Allory and A Londoño-Vallejo. The Genomics platform and NanoString technology of Institut Curie were set up with the support of Agence Nationale de la Recherche (LabEx and EquipEx: ANR-10-IDEX-0001-02 PSL, ANR-11-LBX-0044), Insitut National du Cancer (INCa-DGOS-4654, SIRIC11-002). RNA-seq efforts were supported by a grant from the ICGex program at Institut Curie to A Londoño-Vallejo and A Morillon and benefited from the facilities and expertise of the Next Generation Sequencing platform of Institut Curie, supported by Agence Nationale de la Recherche (ANR-10-EQPX-03, ANR10-INBS-09-08) and Canceropôle Ile-de-France. M Descrimes, M Gabriel, A Morillon, M Pinskaya, and Z Saci were supported by Agence Nationale de la Recherche (DNA-Life) and the European Research Council (ERC-consolidator DARK-616180-ERC-2014) attributed to A Morillon; D Gautheret and HTN Nguyen were supported by ITMO Cancer–Systems Biology (bio2014-04) and Agence Nationale de la Recherche "France Génomique" (ANR-10-INBS-0009) attributed to D Gautheret.

### Authors Contributions

M Pinskaya: supervision, validation, investigation, methodology, project administration, and writing—original draft, review, and editing.
Z Saci: software and formal analysis.
M Gallopin: formal analysis, supervision, and writing—original draft, review, and editing.
M Gabriel: software and formal analysis.
HTN Nguyen: formal analysis.
V Firlej: resources and data curation.
M Descrimes: formal analysis.
A Rapinat: investigation.
D Gentien: investigation.
A De la Taille: resources and data curation.
A Londoño-Vallejo: data curation.
Y Allory: data curation.
D Gautheret: conceptualization, formal analysis, supervision, funding acquisition, project administration, and writing—original draft, review, and editing.

A Morillon: conceptualization, supervision, funding acquisition, project administration, and writing—original draft, review, and editing.

## Conflict of Interest Statement

The authors declare that they have no conflict of interest.

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
