## [Reviewer comments · Life Science Alliance]

Life Science Alliance

Reference-free transcriptome exploration reveals novel RNAs for prostate cancer diagnosis

Marina Pinskaya, Zohra Saci, Melina Gallopin, Marc Gabriel, Ngoc-Ha Nguyen, Virginie Firlej, Marc Descrimes, Audrey Rapinat, David Gentien, Alexandre De la Taille, Arturo Londono-Vallejo, Yves Allory, Daniel Gautheret, and Antonin Morillon

DOI: <https://doi.org/10.26508/lsa.201900449>

Corresponding author(s): Antonin Morillon, CNRS-Institut Curie, Université PSL, Sorbonne Université and Daniel Gautheret, Université Paris-Sud Orsay

Review Timeline:

Submission Date:	2019-06-04
Editorial Decision:	2019-07-19
Revision Received:	2019-10-02
Editorial Decision:	2019-10-29
Revision Received:	2019-11-05
Accepted:	2019-11-05

Scientific Editor: Andrea Leibfried

Transaction Report:

July 19, 2019

Re: Life Science Alliance manuscript #LSA-2019-00449

Dr. Antonin Morillon
CNRS-institut Curie
UMR3244
26, rue d'Ulm
Paris 75005
France

Dear Dr. Morillon,

Thank you for submitting your manuscript entitled "Reference-free transcriptome exploration reveals novel RNAs for prostate cancer diagnosis" to Life Science Alliance. The manuscript was assessed by expert reviewers, whose comments are appended to this letter.

As you will see, the reviewers appreciate your work and provide constructive input on how to further strengthen it. We would thus like to invite you to submit a revised version of your manuscript to us, addressing the individual concerns raised by the reviewers. Importantly, code and data tables need to get included, more evidence to support your conclusions is required and more information needs to get provided (rev#1 and #3). Validation should get provided or lack thereof discussed (rev#3), and the superiority of the DEkupi-based approach needs to get demonstrated (rev#3).

Thank you for this interesting contribution to Life Science Alliance. We are looking forward to receiving your revised manuscript.

Sincerely,

B. MANUSCRIPT ORGANIZATION AND FORMATTING:

Reviewer #1 (Comments to the Authors (Required)):

The paper describes a method for the discovery of potential novel gene expression-based biomarkers. Moreover, the authors performed a proof of principle and identified potential markers for

prostate cancer diagnostics and tested them across multiple datasets.

The authors identified differentially represented kmers using DE-kupl between tumor and control samples, they assembled the kmers to contigs, and annotated them by mapping the contigs to existing annotations. It is unclear to me why the authors have had to develop a de novo transcriptome assembler when multiple well-tested assemblers are available. Also, I would require the authors to provide the code and data tables as prerequisite to publication and not have it provided only upon request.

Generally, the manuscript is well written and I think that the authors should be able to address the comments below.

Major points:

There is insufficient evidence that the newly detected markers are not part of known transcripts and were assembled into shorter contigs because of the approach the authors used for assembly. For example, if I interpret Fig S3B correctly, there is no expression in the minus strand in the FBP2 locus. If so, P2 expression could represent part of the PCAT7 transcript. This is important since it is unclear whether the newly reported transcripts are truly standalone transcripts.

There is insufficient description of the validation using the TCGA data. These data, as the authors note, is non-strand specific and it is based on polyA-enriched RNA. Therefore, a thorough description of the approach used to estimate the expression of the short contigs that the authors have identified is required. Moreover, comparison to randomly selected randomized contigs that span similar transcripts/genomic regions is advisable.

Page 18: The results for P2 and P6 are particularly interesting, yet do the authors have evidence that parts of the PCAT7 that do not overlap with P2 and P6 are not overexpressed?

Minor points:

Page 5: "The only example of an RNA-based biomarker so far introduced in clinical practice is the PCA3 lncRNA in prostate cancer (PCa)": is that correct? Isn't the panel PAM50 used in the clinic as well?

Page 6: "impossible for co-expressed paired sense/antisense transcripts": this is true only for non-stranded RNAseq data and when overlapping gene models on opposite strands do not exist

Page 6: "the most potent subset of contigs and" the term potent is vague in this sentence, the wording needs to be clearer.

Page 6: "alternative NanoString assay" change alternative to custom
Did DE-kupl detect known diagnostic markers, such as pca3?

Page 9: "...which is deleterious for splice graph-based assemblers..." What is the meaning of the word deleterious in this context?

Page 16: "Expression of DE-kupl contigs is independent on tumor risk and recurrence Metrics": change on to of

Reviewer #2 (Comments to the Authors (Required)):

The manuscript describes a novel method to identify potential novel ncRNA-based biomarkers identified by RNA-sequencing technologies.

The proposed computational method shows high sensitivity, greater clinical performance than PCA3 in the discrimination of high-risk tumors. Furthermore, this method has the potential to be applied also to other cancer type and pathology other than prostate cancer.

The manuscript is interesting, well written and organized in a very rational sequence. This makes it very straightforward to follow the general rationale and description of the results.

At this point, no major weaknesses or issues are identified. Therefore, it is deemed ready for publication.

Reviewer #3 (Comments to the Authors (Required)):

The manuscript of Pinskaya et al. concerns the mining of RNA-Seq data for unannotated transcripts that may serve as clinical biomarker. The authors have previously published a method called "DE-kupl" to find sequences that are enriched among the RNA-Seq reads of one group of samples relative to another (differential expression of k-mers), in a manner that does not rely on a reference genome or transcriptome. To demonstrate, as a proof of principle, the value of this approach for biomarker discovery, they apply it to RNA-Seq data from prostate biopsy samples, identify a potential cancer signature and validate it in an independent cohort.

The novelty lies in the observation that typical pipelines only use and consider annotated genomic features and hence may miss yet unknown transcripts, particularly long non-coding RNAs. They succeed in discovering novel lncRNAs with specific enrichment in cancer samples over healthy ones. This makes this paper a valuable contribution, as it convincingly demonstrates that this approach is powerful, practical and complementary to established methods. Nevertheless, it should be clear that this paper is a proof of principle, and does not go all the way of establishing a new cancer signature. It does not claim to do so, but needs some clarifications in this area, as discussed below.

Major issues:

1. The paper establishes two "signatures" (one using only novel lncRNAs and one mixing novel and known transcripts). The term "signature" is a bit vague: does it denote merely a list of genes, or an actual formula, how the expression values for these genes should be combined (typically as a weighted sum) to yield a score or a 'cancer yes/no' call? Here, it seems that the authors use their Discovery and Selection Cohorts only to achieve the former: they find the coefficients for weighting the signature genes by applying logistic regression to their Validation Cohort. A proper validation, however, would be to use the Discovery and Selection Data to fully establish a decision rule (i.e., something like: Measure these genes, multiply them with these coefficients, add them up to get score, call "cancer" or "not cancer" if the score is above this threshold.) Then, test this rule on the validation cohort.

Using the present data, this might, unfortunately, not be possible, because the TCGA data is polyA-selected and not stranded. As the authors argue themselves, this makes this data quite

suboptimal, and using regression coefficients established from the stranded, total-RNA data of the Discovery and Selection Cohorts is unlikely to work well.

It is hence somewhat understandable that the authors did the next best thing of using some form of cross validation to both train and validate a logistic regression classifier on the Validation Set. Nevertheless, this limitation should be acknowledged more clearly, and the Methods parts need to describe the used cross validation scheme (nested, I hope) in much more detail. At the moment, it only lists other tools and the reader would have to read their manuals to understand what CV might have been employed.

As all this lessens the support that Fig. 6B gives to the authors overall claims, the core result is hence rather Figure 4. I hence would like to make clear that I consider Figure 4 sufficient to prove that main claim, namely that looking for yet unknown lncRNAs using the authors' pipeline does allow to find promising potential biomarkers. Establishing how well the specific markers work that were found here in this proof of principle is what Figure 6 tries to do, but doing so properly would, of course, require a new clinical study applying the NanoString assay to a new independent cohort, and that is of course not something we should expect here. I only feel that the limitations of the validation need to be discussed more clearly.

Finally, it should also be acknowledged that the literature claims that PCA3 is a good biomarker because it can be detected in urine and is predictive there, while the present manuscript only discusses assays of biopsies.

2. The authors remove ("mask") all known coding genes, in order to focus on new lncRNAs. This is useful to demonstrate DE-kupl. However, to be able to argue that the DEkupl-based approach is superior to normal reference-based approaches, the authors need to show that a standard analysis yields less useful results in terms of potential markers. The authors to compare the DE-kupl approach with a reference-based one (which they term "HoLdUp"), but there, too, they restrict themselves to lncRNAs. I suppose that there are less useful differences in coding genes, but this should be shown. Or, if there are many differences, the argument, why non-coding transcripts are more useful needs to be argued more explicitly? (Isn't the special feature of PCA3 that it gets secreted via exosomes? If so, are the other transcripts a prostate cancer might secrete also likely to be predominantly non-coding?)

Further issues and questions:

3. How does the lack of strandedness of the TCGA data affect the quantification of those selected transcripts that are antisense to coding genes? Do the coding genes contribute to the counts? Or are the k-mers all antisense to introns?

4. In all box plot figures, y axes are labelled "Log10(counts)". However, their values descend below 0. What would a count of 0.1 reads mean? Are these maybe counts divided by some normalisation constant?

5. Discovery and Selection Set are taken from the same cohort. It should hence be stated explicitly whether they overlap.

6. What is a "normal sample"? A sample taken from in biopsy of a patient who then turned out to

not suffer from prostate cancer? Or a sample from a healthy region of a tumour-bearing prostate? If the former, this would be unsatisfactory, if the latter, the PAIR cohort should not be described as a cohort of "prostate cancer patients" but as one of patients tested for prostate cancer.

7. The differences in p value ranking does not indicate "remarkable discrepancies" between the cohorts (p. 16). p value rankings rarely agree between cohort of different sizes due to the effect of differences in statistical power.

8. How were the counts normalized? Simply by dividing by the total read count? Also in the NanoString data?

9. It is nice that the R code is included in Suppl File S15. It would be even better if the input files were provided. Without the I could not run and hence not check the code.

Minor points:

10. How do the Ving-generated figures (e.g., Fig 3D) aggregate samples? Are these coverages of single samples or sums over all samples in a group?

11. misspelling "quartile" instead of "quantile" twice

12. page 12: "contigs ... were filtered out. We also retained ...". "Filtered out" is supposed to mean "not retained", right? The "also" is confusing.

13. Version of DE-kupl is stated as "(June 2017)". Is this version 1.0?

14. The manuscript correctly cites the bioinformatics papers for most tools they have used. Two are missing, though: featureCounts and JellyFish.

Reviewer #1 (Comments to the Authors (Required)):

The paper describes a method for the discovery of potential novel gene expression-based biomarkers. Moreover, the authors performed a proof of principle and identified potential markers for prostate cancer diagnostics and tested them across multiple datasets.

The authors identified differentially represented kmers using DE-kupl between tumor and control samples, they assembled the kmers to contigs, and annotated them by mapping the contigs to existing annotations. It is unclear to me why the authors have had to develop a de novo transcriptome assembler when multiple well-tested assemblers are available. Also, I would require the authors to provide the code and data tables as prerequisite to publication and not have it provided only upon request.

In this work, we used total stranded RNA-seq for transcriptome analysis and transcripts discovery and quantification in clinical samples. These datasets are much more complex since enriched in intronic reads, but also of lower coverage for the same number of total reads than poly(A)-selected RNA-seq. This renders classic computational analysis, used by the majority of researchers, much more laborious with high fraction of false positive hits and long manual curating.

Indeed, the existing assembler pipelines are Cufflink based, highly consuming in time and memory rendering computational analysis specifically long when dealing with multiple total stranded RNA-seq datasets. Our de novo transcriptome assembler HoLdUp has been developed to allow much faster and easier processing of total stranded RNA-seq datasets allowing assembly of transcription unit contigs without proper intron/exon annotation. Regardless the lack of precision in reconstruction of transcript isoforms, HoLdUp gives us rapid but fundamental information on the whole transcription unit. Such approach fulfilled completely our aims in discovery of novel transcripts allowing further expression counting and differential analysis. Since DE-kupl highlights only RNA subsequences but not the whole transcript, HoLdUp complemented our work allowing to assign some of contigs to de novo assembled transcription units. This can be valuable for further functional studies but dispensable for contraction of contig-based clinical signatures. This issue was more clearly discussed in the Discussion section (page 18).

To complement a search for biomarkers and validate some of those among unannotated mRNA isoforms or novel lncRNAs, further work is required; and in this case the long read RNA-sequencing would be the most appropriate.

The code and data tables were uploaded into GitHub and are accessible upon a link: <https://github.com/MorillonLab/Prostate-kmer-signatures>.

The HoLdUp package with the *readme* file, annotation and count tables can be freely uploaded through the link: http://xfer.curie.fr/get/3isUnBCOhap/HoLdUp_v0.05.zip.

All this information has been included in the manuscript body.

Generally, the manuscript is well written and I think that the authors should be able to address the comments below. Major points:

There is insufficient evidence that the newly detected markers are not part of known transcripts and were assembled into shorter contigs because of the approach the authors used for assembly. For example, if I interpret Fig S3B correctly, there is no expression in the minus strand in the FBP2 locus. If so, P2 expression could represent part of the PCAT7 transcript. This is important since it is unclear whether the newly reported transcripts are truly standalone transcripts.

We thank the reviewer for this remark. Indeed, the DE-kupl pipeline does not perform full-length assembly and detects only subsequences still unreferenced in the existing annotation. To assign them to referenced genes, we analyzed the overlap of DE-kupl contigs with recent GENCODE and MiTranscriptome annotations (page 7). Some contigs are embedded into already annotated genes, but represent yet unreferenced transcript variations, as in case of the contig P6. In contrast, contig P2 maps downstream the existing PCAT7 GENCODE annotation. Considering uniquely RNA-seq reads profile and exon junction mapping, we hypothesized that this contig represents an extension of the last exon. For 100% certainty, 3'-RACE or alternative experiments are required to correct the existing PCAT7 gene annotation. This issue, regardless of its importance, was not in the scope of this work but we have put more efforts to explain the case (pages 9-10) with additional Sashimi plot of PCAT7 for better illustration of our hypothesis (Supplemental Fig. S3D).

There is insufficient description of the validation using the TCGA data. These data, as the authors note, is non-strand specific and it is based on polyA-enriched RNA. Therefore, a thorough description of the approach used to estimate the expression of the short contigs that the authors have identified is required. Moreover, comparison

to randomly selected randomized contigs that span similar transcripts/genomic regions is advisable.

We thank the reviewer for this advice. Indeed, TCGA data are of a low quality for nowadays standards. To clarify how we process the data, detailed description of the counting procedure was added to Material & Methods "Variable selection using the LASSO penalized logistic regression and external validation of signatures" (page 23). In addition, we added an alternative counting procedure for TCGA datasets, in which contigs expression has been quantified using randomly selected k-mers. Both quantifications showed high concordance and the results were presented in Supplemental Figure S8.

Page 18: The results for P2 and P6 are particularly interesting, yet do the authors have evidence that parts of the PCAT7 that do not overlap with P2 and P6 are not overexpressed?

This remark joins in part the abovementioned point. The case of PCAT7 is now discussed in more details in the Results section (pages 9-10) including the differential expression analysis (Figure S3B). Indeed and in accordance with other publications, we confirmed here that the whole PCAT7 gene is overexpressed in PCa specimens comparing to normal tissues (DESeq, FC > 10, p-value < 6,3E-20). However, in this work we were particularly interested in yet unannotated variants of PCAT7 and for this reason we selected two DE-kupl contigs for further NanoString and TCGA validation: P6 within the existing PCAT7 gene annotation, P2 as the best DE contig antisense to the FBP2 gene but out of the existing annotation. Again further isoforms definitions-dedicated experiments would clearly answer this question but again, remain out of the scope of this manuscript. Of note, since the whole gene is DE, many other subsequences (DE-kupl contigs) are also DE (Table S2) but we selected the most significant hit, P6.

Minor points:

Page 5: "The only example of an RNA-based biomarker so far introduced in clinical practice is the PCA3 lncRNA in prostate cancer (PCa)": is that correct? Isn't the panel PAM50 used in the clinic as well?

In this work, we focused on prostate cancer and lncRNAs, so the sentence was changed accordingly (page 5): "The only example of a lncRNA-based biomarker so far introduced in clinical practice is the PCA3 lncRNA in prostate cancer (PCa)". The PAM50 signature is based on quantification of mRNAs and is used for breast cancer prognosis. For this reason we didn't mentioned it in the text, but indeed RNA signatures are already in use in clinics including PCa diagnosis and it was acknowledged in discussion part (page 17).

Page 6: "impossible for co-expressed paired sense/antisense transcripts": this is true only for non-stranded RNAseq data and when overlapping gene models on opposite strands do not exist

The reviewer is correct, the sentence was changed : " for non-stranded RNA-seq reads counting is less accurate at 5' RNA ends or even impossible for co-expressed paired sense/antisense transcripts and for yet unannotated RNAs among noncoding, fusion, repeat-derived transcripts"

Page 6: "the most potent subset of contigs and" the term potent is vague in this sentence, the wording needs to be clearer.

We changed it for "the subset of contigs with best differential expression features".

Page 6: "alternative NanoString assay" change alternative to custom. Did DE-kupl detect known diagnostic markers, such as pca3?

"alternative" has been changed to "custom".

Of note, DE-kupl detects several PCA3 subsequences (Supplemental Table S2), all of them represent yet unreferenced transcript variants. Aiming to find new biomarkers, we did not select these contigs on purpose.

Page 9: "...which is deleterious for splice graph-based assemblers..." What is the meaning of the word deleterious in this context?

The word "deleterious" has been replaced by "laborious"

Page 16: "Expression of DE-kupl contigs is independent on tumor risk and recurrence Metrics": change on to of

done

Reviewer #2 (Comments to the Authors (Required)):

The manuscript describes a novel method to identify potential novel ncRNA-based biomarkers identified by RNA-sequencing technologies.
The proposed computational method shows high sensitivity, greater clinical performance than PCA3 in the discrimination of high-risk tumors. Furthermore, this method has the potential to be applied also to other cancer type and pathology other than prostate cancer.
The manuscript is interesting, well written and organized in a very rational sequence. This makes it very straightforward to follow the general rationale and description of the results.
At this point, no major weaknesses or issues are identified. Therefore, it is deemed ready for publication.

We would like to thank the reviewer for recognizing the quality and the value of our work. We hope also it can be diffused now to the community since we believe it is of general interest both in methodology aspects but also in defining novel cancer diagnosis strategies.

Reviewer #3 (Comments to the Authors (Required)):

The manuscript of Pinskaya et al. concerns the mining of RNA-Seq data for unannotated transcripts that may serve as clinical biomarker. The authors have previously published a method called "DE-kupl" to find sequences that are enriched among the RNA-Seq reads of one group of samples relative to another (differential expression of k-mers), in manner that does not rely on a reference genome or transcriptome. To demonstrate, as a proof of principle, the value of this approach for biomarker discovery, they apply it to RNA-Seq data from prostate biopsy samples, identify a potential cancer signature and validate it in an independent cohort.

The novelty lies in the observation that typical pipelines only use consider annotated genomic features and hence may miss yet unknown transcripts, particularly long non-coding RNAs. They succeed in discovering novel lncRNAs with specific enrichment in cancer samples over healthy ones. This makes this paper a valuable contribution, as it convincingly demonstrates that this approach is powerful, practical and complementary to established methods. Nevertheless, it should be clear that this paper is a proof of principle, and does not go all the way of establishing a new cancer signature. It does not claim to do so, but needs some clarifications in this area, as discussed below.

We would like to thank the referee to have perfectly understood the value of our manuscript and acknowledge it as a valuable contribution. We perfectly agree that it should not be misunderstood that our results represent a proof a principle and not a clinical statement. We have answered and modified the manuscript accordingly.

Major issues:

1. The paper establishes two "signatures" (one using only novel lncRNAs and one mixing novel and known transcripts). The term "signature" is bit vague: does it denote merely a list of genes, or an actual formula, how the expression values for these genes should be combined (typically as a weighted sum) to yield a score or a 'cancer yes/no' call? Here, it seems that the authors use their Discover and Selection Cohorts only to achieve the former: they find the coefficients for weighting the signature genes by applying logistic regression to their Validation Cohort. A proper validation, however, would be to use the Discovery and Selection Data to fully establish a decision rule (i.e., something like: Measure these genes, multiply them with these coefficients, add them up to get score, call "cancer" or "not cancer" if the score is above this threshold.) Then, test this rule on the validation cohort.

Using the present data, this might, unfortunately, not be possible, because the TCGA data is poly(A)-selected and not stranded. As the authors argue themselves, this makes this data quite suboptimal, and using regression coefficients established from the stranded, total-RNA data of the Discovery and Selection Cohorts is unlikely to work well.

We thank the reviewer for this comment. To avoid ambiguity, we introduced a sentence explaining the usage of "signature" in this work (page 9): "Hereafter, we will use the term *signature* to describe the set of contigs or genes selected for their ability to predict a sample status".

Indeed, in our case, we define a set of contigs/genes to predict a sample status, but this does not correspond to a signature with weight coefficients. We had to compute coefficients on the Validation set since the nature of libraries in the Discovery and Validation sets are different. As acknowledged by reviewer, keeping the coefficients

would not work well. Hence, we now discuss this limitation in the text, Materials & Methods section, "Variable selection using the LASSO penalized logistic regression and external validation of signatures" (page 23).

It is hence somewhat understandable that the authors did the next best thing of using some form of cross validation to both train and validate a logistic regression classifier on the Validation Set. Nevertheless, this limitation should be acknowledged more clearly, and the Methods parts need to describe the used cross validation scheme (nested, I hope) in much more detail. At the moment, it only lists other tools and the reader would have to read their manuals to understand what CV might have been employed.

The reviewer is right; we now detailed the cross-validation scheme in the Methods section (page 23). Of note, here the classifier selection is of less importance since we are more interested in comparing the performance of different subsets of genes/contigs between each other than achieving the best possible prediction.

As all this lessens the support that Fig. 6B gives to the authors overall claims, the core result is hence rather Figure 4. I hence would like to make clear that I consider Figure 4 sufficient to prove that main claim, namely that looking for yet unknown lncRNAs using the authors' pipeline does allow to find promising potential biomarkers. Establishing how well the specific markers work that were found here in this proof of principle is what Figure 6 tries to do, but doing so properly would, of course, require a new clinical study applying the NanoString assay to a new independent cohort, and that is of course not something we should expect here. I only feel that the limitations of the validation need to be discussed more clearly.

We agree with the reviewer that Figure 4 supports the main claim of the paper. However, ROC analysis is the most commonly used test in clinical research giving indexes of accuracy (AUC) as a meaningful interpretation for disease classification from healthy subjects (Linden 2006). To complete the proof of concept and to render our results easier to interpret and appealing for medical researches and comparable to other studies (Ploussard and de la Taille 2010), (Xu et al. 2018), we therefore performed ROC analysis and included them in Figure 6.

As further proposed by the reviewer, a new independent clinical study is required for more robust validation including a new cohort but also an alternative RNA quantification technique more applicable in clinics (NanoString, digital PCR, others). This point was introduced in the Results section (page 15).

Finally, it should also be acknowledged that the literature claims that PCA3 is a good biomarker because it can be detected in urine and is predictive there, while the present manuscript only discusses assays of biopsies.

We totally agree with the reviewer. PCA3, as well as another RNA-based clinical test Decipher have been acknowledged in Discussion section (page 25). The usage of DE-kupl contigs as urine biomarkers is now part of an on-going project in the lab. According to our preliminary results (unpublished), the development of a urine signature requires the whole discovery pipeline to be applied directly to RNA-seq from urine samples, using other cohorts and RNA measurements for selection and validation. This issue is discussed at the end of the manuscript (page 19).

2. The authors remove ("mask") all known coding genes, in order to focus on new lncRNAs. This is useful to demonstrate DE-kupl. However, to be able to argue that the DEkupl-based approach is superior to normal reference-based approaches, the

authors need to show that a standard analysis yields less useful results in terms of potential markers. The authors to compare the DE-kupl approach with a reference-based one (which they term "HoLdUp"), but there, too, they restrict themselves to lncRNAs. I suppose that there are less useful differences in coding genes, but this should be shown. Or, if there are many differences, the argument, why non-coding transcripts are more useful needs to be argued more explicitly? (Isn't the special feature of PCA3 that is gets secreted via exosomes? If so, are the other transcripts a prostate cancer might secrete also likely to be predominantly non-coding?)

We thank the reviewer for this remark. To compare our approach to a standard one, we performed a conventional analysis using GENCODE annotated genes, coding and noncoding. A set of 520 up-regulated genes was selected in the Discovery cohort by DEseq2 and used for Lasso logistic regression. A retrieved signature of 5 genes (whole gene count, WGC) was then tested for prediction accuracy in TCGA (Validation Set) (page 23). The result was reported in the Results section (page 14-15, Fig. S7B). Strikingly, the signature derived from 520 DE genes performed as well as the k-mer based retrieved from 22 or 15 probes for mixed and new-lnc RNA signatures, respectively. This result reinforces the power of the k-mer approach, which requires much less sequence combinations to achieve comparable performance as for annotated genes. In addition, this result was obtained using the low-coverage unstranded polyA+ datasets of TCGA which are more enriched for highly expressed poly(A)+ genes, than non poly(A) and low abundant RNA isoforms (antisense, circular RNA, splicing variants etc). The fact that TCGA-based quantification of k-mer contigs discovered from the total stranded RNA-seq is largely disfavored further assures the potential of k-mer based signatures for biomarkers discovery. We discussed this issue more in detail in Results part (page 15).

Regardless rather high performance of the gene-based signature, here we tuned our discovery workflow towards yet unannotated subsequences within noncoding regions for several reasons: 1) lncRNA are claimed to have more specific expression than mRNAs and hence can potentially yield more specific and robust biomarkers; 2) the only lncRNA-based prostate biomarker used so far in clinics is PCA3 and we used it as a benchmark for comparison with our lncRNA signature; 3) we aimed to identify novel lncRNA candidates associated with prostate cancer since our total RNA-seq discovery dataset had a high potential for the discovery of unannotated transcripts. Moreover, lncRNAs including PCA3 were also detected in biological fluids opening a possibility of their usage in non-invasive clinical tests (Introduction, page 4).

Further issues and questions:

3. How does the lack of strandedness of the TCGA data affect the quantification of those selected transcripts that are antisense to coding genes? Do the coding genes contribute to the counts? Or are the k-mers all antisense to introns?

First, not all antisense contigs reside in introns, but by performing a manual selection of contigs we tried to avoid the regions containing reads in both strands in the total stranded RNA-seq. Second, we checked the expression counts depending on contig type (antisense or intergenic) and didn't find any bias. This was mentioned at page 16: "Detection of these contigs in TCGA-PRAD was compromised independently of their genomic location (intergenic or antisense) or of the expression level of a sense-paired gene."

4. In all box plot figures, y axes are labelled "Log10(counts)". However, their values descend below 0. What would a count of 0.1 reads mean? Are these maybe counts divided by some normalization constant?

We thank the reviewer for this remark. Indeed, "counts" correspond to normalized values. The legends of corresponding figures were corrected accordingly.

In NanoString (Figure 4A, Figure 5), the data were normalized through the use of NanoString's intrinsic negative and positive controls according to the normalization approach of the nSolver analysis software and then contig expression was calculated relative to the average signal of three housekeeping genes (*GPATCH3*, *ZNF2* and *ZNF346*). In RNA-seq gene expression was normalized using the DEseq2 median of ratio (Anders and Hubers, 2010) (Figure 4B, S5). Contigs expression from DE-kupl have been normalized using the median of ratio normalization method. These specifications were introduced in Methods section.

5. Discovery and Selection Set are taken from the same cohort. It should hence be stated explicitly whether they overlap.

The text was added (page 10):

"This cohort contained one additional specimen for normal tissue and 119 additional tumor specimens".

6. What is a "normal sample"? A sample taken from in biopsy of a patient who then turned out to not suffer from prostate cancer? Or a sample from a healthy region of a tumour-bearing prostate? If the former, this would be unsatisfactory, if the latter, the PAIR cohort should not be described as a cohort of "prostate cancer patients" but as one of patients tested for prostate cancer.

The majority, if not all, clinical cohorts including TCGA and our PAIR cohort contain normal tissues coming from the surrounding non-cancerous area of patients diagnosed for Prostate Cancer and subjected to prostatectomy.

7. The differences in p value ranking does not indicate "remarkable discrepancies" between the cohorts (p. 16). p value rankings rarely agree between cohort of different sizes due to the effect of differences in statistical power.

We agree with the reviewer on the point of a statistical power dependence of the cohort size. The word "remarkable" was deleted from the text and "cohort size" was added as an argument that may explain observed differences. However, given the fact that the Discovery and Selection sets differ a lot in sample size but still have comparable ranking, we cannot exclude that differences also come from other parameters (cohort origin/sample preparations, RNA-seq and counting procedures).

8. How were the counts normalized? Simply by dividing by the total read count? Also in the NanoString data?

NanoString expression measurements were subjected to double normalization: one is intrinsic to the assay and uses internal negative and positive controls; second is performed afterwards relative to three lowly expressed housekeeping genes which we selected as stably expressed. Contigs expression from DE-kupl have been normalized using the median of ratio normalization method. In RNA-seq datasets gene expression have been normalized using the DEseq2 median of ratio (Anders and Hubers, 2010). Contigs expression from DE-kupl have been normalized using the median of ratio normalization method. This was mentioned in the Material section.

9. *It is nice that the R code is included in Suppl File S15. It would be even better if the input files were provided. Without it I could not run and hence not check the code.*

Now all data and codes are uploaded to a GitHub repository:
<https://github.com/MorillonLab/Prostate-kmer-signatures>

Minor points:

10. *How do the Ving-generated figures (e.g., Fig 3D) aggregate samples? Are these coverages of single samples or sums over all samples in a group?*

This point was clarified on page 30:

RNA-seq reads profiling along a locus of interest was performed using our published software VING using one “normal” and one “tumor” RNA-seq subsets build by random sampling of 10% of reads from each raw data sample”.

11. *misspelling "quartile" instead of "quantile" twice*

We used “quartile” as a subtype of quantile since it splits off the lowest 25% of data from the highest 75% - this is exactly the threshold we used for expression strength filter.

12. *page 12: "contigs ... were filtered out. We also retained ... ". "Filtered out" is supposed to mean "not retained", right? The "also" is confusing.*

“also” has been removed

13. *Version of DE-kupl is stated as "(June 2017)". Is this version 1.0?*

No, but the versions are ordered by the date of release. Hence, the date will allow retrieving the corresponding DE-kupl code from the GitHub repository.

14. *The manuscript correctly cites the bioinformatics papers for most tools they have used. Two are missing, though: featureCounts and JellyFish.*

Two citations were added : FeatureCounts (PMID 24227677), Jellyfish (PMID 21217122)

October 29, 2019

RE: Life Science Alliance Manuscript #LSA-2019-00449RR

Dr. Antonin Morillon
CNRS-Institut Curie, Université PSL, Sorbonne Université
UMR3244
26, rue d'Ulm
Paris 75005
France

Dear Dr. Morillon,

Thank you for submitting your revised manuscript entitled "Reference-free transcriptome exploration reveals novel RNAs for prostate cancer diagnosis".

As you will see, reviewer #3 re-assessed your revised version and now supports publication, pending further text changes to openly state the limitations of the approach in determining biomarkers. We agree that the requested discussion and text changes are warranted and of benefit to the community, so please introduce them. Furthermore, please:

- add a callout to Fig S6A
- please move the data currently hosted on the curie server to a stable depository

Once we receive such a further revised version, we will swiftly move towards acceptance and production of your paper.

A. FINAL FILES:

-- High-resolution figure, supplementary figure and video files uploaded as individual files: See our detailed guidelines for preparing your production-ready images, <http://www.life-science->

alliance.org/authors

B. MANUSCRIPT ORGANIZATION AND FORMATTING:

Sincerely,

Andrea Leibfried, PhD
Executive Editor
Life Science Alliance
Meyershofstr. 1
69117 Heidelberg, Germany
t +49 6221 8891 502
e a.leibfried@life-science-alliance.org

Reviewer #3 (Comments to the Authors (Required)):

The authors have provides suitable replies to the issues I raised in my review. All in all, I would say that this is now a fine paper and ready for publication in most parts.

There is, however, one issue left that I still feel is very misleading. It is related to the issue of "signatures" and ROC curves that I had raised in my previous report, where I asked that the limitations of the validation need to be discussed more explicitly.

I briefly review, for the Editor's benefit, the usual procedure in finding biomarkers: One uses the discovery data set to come up with a rule that allows one to take expression values from a new sample and calculate a score, such that high values of the score indicate that the sample is likely a cancer sample and low values that it is not. Strictly speaking, the threshold, i.e., the score value, where one switches from calling "not cancer" to calling "cancer" needs to be found at this stage, too. Typically, such a decision procedure has the form of a list of genes with corresponding numbers, called coefficients. To obtain the score for a new sample, the rule is then usually to multiply the expression of each of the genes in the list with the specified coefficient, and add up all these values to obtain the score.

Validation now means to take one sample at a time from some independent validation cohort, calculate the score and decide whether the sample is cancer or not. Obviously, this needs to be done without using the information whether the sample really is from a cancer patient or not. Only then, one can afterwards check how many of the cancer-or-not calls were correct and thus calculate sensitivity or specificity of the procedure that one is proposing. Redoing the last step for different decision thresholds allows one to get a ROC curve.

Using the discovery cohort, the authors have obtained only a list of genes, but not the coefficients. Therefore, they could not calculate a score for a sample from the validation cohort. Rather, they have used the validation cohort to find the coefficients. They needed to do so because their discovery and validation set have been produced using quite different lab procedures, and therefore, coefficients from one set would not work on the other set. Unfortunately, to get the coefficients for the validation set, they needed to use the information on which of the validation samples were cancer samples and which were not. Using the same data to train a classifier (obtain the coefficients) and to test (validate) it is not allowed, because it will show a much too optimistic assessment of the classifier's performance. The authors addressed this issue by performing leave-one-out cross validation, i.e., always training the classifiers on all sample but one and testing it on the left out one. This is considered an acceptable interim solution in the field while developing a method, but is considered insufficient to substantiate any claim of actually providing a new biomarker signature. For that, a validation an an entirely independent cohort is required

Hence, we authors cannot make a robust statement about how good their lncRNA signature really is. If the purpose of the present paper were to establish a new biomarker set to detect prostate cancer, this would not be acceptable. One would have to insist that the authors somehow obtain a proper validation data set.

However, the purpose of the paper is to demonstrate the potential of using the authors' DE-kupl

method, and the prostate cancer task is a mere proof of principle. I feel that the internal leave-one-out cross validation is sufficient for that purpose. After all, this already shows that the selection of lncRNA performed in the first cohort was useful, and more useful than a selection of mRNA, and this is all that the authors wish to claim.

A reader familiar with machine learning will understand that the mention of "leave-one-out cross validation" implies all the limitations I explained above. Most readers of LSA, however, will not know about these intricacies. They will see the ROC curve in Figure 6, compare it with similar ROC curves in other publications, and assume that the method presented here is a better biomarker than the established ones. The Discussion hence has to make very clear that this claim cannot be made.

I appreciate that there are many papers out that are misleading in precisely the described sense. Incorrect validation is probably the single most common mistake in this branch of literature, and doing so properly can even set a paper at a disadvantage as it looks weaker. And, to be clear, nothing that the authors claim is actually wrong or dishonest; it is merely written in a way that would hide the limitations of the result from the non-expert reader. I would like to suggest that the authors edit their Discussion to state clearly that the information spill-over from discovery to validation set precludes them from calculating objective performance measures of the obtained "signature". And as I understand that I am here suggesting to be more "honesty" than many papers in this field are, I would encourage the Editor to clearly state that openly discussing such limitations will not be considered a negative.

Dear Dr. Leibfried,

We are very grateful for attentive revising of the manuscript "Reference-free transcriptome exploration reveals novel RNAs for prostate cancer diagnosis".

We introduced the changes into the text accordingly:

- add a callout to Fig S6A

Done

- please move the data currently hosted on the curie server to a stable depository

The data have been transferred to GitHub:

https://github.com/MorillonLab/HoLDuP_pipeline.

The link was added in Materials and Methods (page 17)

Reviewer #3 (Comments to the Authors (Required)):

The authors have provided suitable replies to the issues I raised in my review. All in all, I would say that this is now a fine paper and ready for publication in most parts.

There is, however, one issue left that I still feel is very misleading. It is related to the issue of "signatures" and ROC curves that I had raised in my previous report, where I asked that the limitations of the validation need to be discussed more explicitly.

I briefly review, for the Editor's benefit, the usual procedure in finding biomarkers: One uses the discovery data set to come up with a rule that allows one to take expression values from a new sample and calculate a score, such that high values of the score indicate that the sample is likely a cancer sample and low values that it is not. Strictly speaking, the threshold, i.e., the score value, where one switches from calling "not cancer" to calling "cancer" needs to be found at this stage, too. Typically, such a decision procedure has the form of a list of genes with corresponding numbers, called coefficients. To obtain the score for a new sample, the rule is then usually to multiply the expression of each of the genes in the list with the specified coefficient, and add up all these values to obtain the score.

Validation now means to take one sample at a time from some independent validation cohort, calculate the score and decide whether the sample is cancer or not. Obviously, this needs to be done without using the information whether the sample really is from a cancer patient or not. Only then, one can afterwards check how many of the cancer-or-not calls were correct and thus calculate sensitivity or specificity of the procedure that one is proposing. Redoing the last step for different decision thresholds allows one to get a ROC curve.

Using the discovery cohort, the authors have obtained only a list of genes, but not the coefficients. Therefore, they could not calculate a score for a sample from the validation cohort. Rather, they have used the validation cohort to find the coefficients. They needed to do so because their discovery and validation set have been produced using quite different lab procedures, and therefore, coefficients from one set would not work on the other set. Unfortunately, to get the coefficients for the validation set, they needed to use the information on which of the validation samples were cancer samples and which were not. Using the same data to train a classifier (obtain the coefficients) and to test (validate) it is not allowed, because it will show a much too optimistic assessment of the classifier's performance. The authors addressed this issue by performing leave-one-out cross validation, i.e., always training the classifiers on all sample but one and testing it on the left out one. This is considered an acceptable interim solution in the field while developing a method, but is considered insufficient to substantiate any claim of actually providing a new biomarker signature. For that, a validation an entirely independent cohort is required.

Hence, we authors cannot make a robust statement about how good their lncRNA signature really is. If the purpose of the present paper were to establish a new biomarker set to detect prostate cancer, this would not be acceptable. One would have to insist that the authors somehow obtain a proper validation data set.

However, the purpose of the paper is to demonstrate the potential of using the authors' DE-kupl method, and the prostate cancer task is a mere proof of principle. I feel that the internal leave-one-out cross validation is sufficient for that purpose. After all, this already shows that the selection of lncRNA performed in the first cohort was useful, and more useful than a selection of mRNA, and this is all that the authors wish to claim.

A reader familiar with machine learning will understand that the mention of "leave-one-out cross validation" implies all the limitations I explained above. Most readers of LSA, however, will not know about these intricacies. They will see the ROC curve in Figure 6, compare it with similar ROC curves in other publications, and assume that the method presented here is a better biomarker than the established ones. The Discussion hence has to make very clear that this claim cannot be made.

I appreciate that there are many papers out that are misleading in precisely the described sense. Incorrect validation is probably the single most common mistake in this branch of literature, and doing so properly can even set a paper at a disadvantage as it looks weaker. And, to be clear, nothing that the authors claim is actually wrong or dishonest; it is merely written in a way that would hide the limitations of the result from the non-expert reader. I would like to suggest that the authors edit their Discussion to state clearly that the information spill-over from discovery to validation set precludes them from calculating objective performance measures of the obtained "signature". And as I understand that I am here suggesting to be more "honesty" than many papers in this field are, I would encourage the Editor to clearly state that openly discussing such limitations will not be considered a negative.

To address the remark of the the reviewer#3, we introduced a following statement into Discussion (page 14):

Still, in this study, to compute signature coefficients, sample information (normal vs. tumor) was used since the extended *Selection* and independent *Validation Sets* employed two different technologies for RNA measurements. This precluded us from calculating the objective performance. An additional cohort using the same NanoString technology as the *Selection Set* should now be tested to explore the clinical potential of the obtained signature.

--

November 5, 2019

RE: Life Science Alliance Manuscript #LSA-2019-00449RRR

Dr. Antonin Morillon
CNRS-Institut Curie, Université PSL, Sorbonne Université
UMR3244
26, rue d'Ulm
Paris 75005
France

Dear Dr. Morillon,

Thank you for submitting your Methods entitled "Reference-free transcriptome exploration reveals novel RNAs for prostate cancer diagnosis". I appreciate the introduced changes and it is a pleasure to let you know that your manuscript is now accepted for publication in Life Science Alliance. Congratulations on this interesting work.

DISTRIBUTION OF MATERIALS:

Again, congratulations on a very nice paper. I hope you found the review process to be constructive and are pleased with how the manuscript was handled editorially. We look forward to future exciting submissions from your lab.

Sincerely,

Andrea Leibfried, PhD
Executive Editor
Life Science Alliance
Meyerohofstr. 1
69117 Heidelberg, Germany
t +49 6221 8891 502
e a.leibfried@life-science-alliance.org
www.life-science-alliance.org